# The Laplacian in RL: Learning Representations with Efficient Approximations

**Yifan Wu**[*]
Carnegie Mellon University
yw4@cs.cmu.edu

**George Tucker**
Google Brain
gjt@google.com

**Ofir Nachum**
Google Brain
ofirnachum@google.com

## Abstract

The smallest eigenvectors of the graph Laplacian are well-known to provide a succinct representation of the geometry of a weighted graph. In reinforcement learning (RL), where the weighted graph may be interpreted as the state transition process induced by a behavior policy acting on the environment, approximating the eigenvectors of the Laplacian provides a promising approach to state representation learning. However, existing methods for performing this approximation are ill-suited in general RL settings for two main reasons: First, they are computationally expensive, often requiring operations on large matrices. Second, these methods lack adequate justification beyond simple, tabular, finite-state settings. In this paper, we present a fully general and scalable method for approximating the eigenvectors of the Laplacian in a model-free RL context. We systematically evaluate our approach and empirically show that it generalizes beyond the tabular, finite-state setting. Even in tabular, finite-state settings, its ability to approximate the eigenvectors outperforms previous proposals. Finally, we show the potential benefits of using a Laplacian representation learned using our method in goal-achieving RL tasks, providing evidence that our technique can be used to significantly improve the performance of an RL agent.

## 1 Introduction

The performance of machine learning methods generally depends on the choice of data representation (Bengio et al., 2013). In reinforcement learning (RL), the choice of state representation may affect generalization (Rafols et al., 2005), exploration (Tang et al., 2017; Pathak et al., 2017), and speed of learning (Dubey et al., 2018). As a motivating example, consider goal-achieving tasks, a class of RL tasks which has recently received significant attention (Andrychowicz et al., 2017; Pong et al., 2018). In such tasks, the agent's task is to achieve a certain configuration in state space; e.g. in Figure 1 the environment is a two-room gridworld and the agent's task is to reach the red cell. A natural reward choice is the negative Euclidean (L2) distance from the goal (e.g., as used in Nachum et al. (2018)). The ability of an RL agent to quickly and successfully solve the task is thus heavily dependent on the representation of the states used to compute the L2 distance. Computing the distance on one-hot (i.e. tabular) representations of the states (equivalent to a *sparse* reward) is most closely aligned with the task's directive. However, such a representation can be disadvantageous for learning speed, as the agent receives the same reward signal for all non-goal cells. One may instead choose to compute the L2 distance on $(x, y)$ representations of the grid cells. This allows the agent to receive a clear signal which encourages it to move to cells closer to the goal. Unfortunately, this representation is agnostic to the environment dynamics, and in cases where the agent's movement is obstructed (e.g. by a wall as in Figure 1), this choice of reward is likely to cause premature convergence to sub-optimal policies unless sophisticated exploration strategies are used. The ideal reward structure would be defined on state representations whose distances roughly correspond to the ability of the agent to reach one state from another. Although there are many suitable such representations, in this paper, we focus on a specific approach based on the graph Laplacian, which is notable for this and several other desirable properties.

---

[*]Work performed while an intern at Google Brain.

For a symmetric weighted graph, the Laplacian is a symmetric matrix with a row and column for each vertex. The $d$ smallest eigenvectors of the Laplacian provide an embedding of each vertex in $\mathbb{R}^d$ which has been found to be especially useful in a variety of applications, such as graph visualization (Koren, 2003), clustering (Ng et al., 2002), and more (Chung & Graham, 1997).

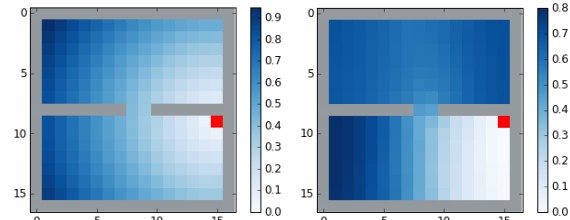

Figure 1: Visualization of the shaped reward defined by the L2 distance from the red cell on an $(x, y)$ representation (left) and Laplacian representation (right).

Naturally, the use of the Laplacian in RL has also attracted attention. In an RL setting, the vertices of the graph are given by the states of the environment. For a specific behavior policy, edges between states are weighted by the probability of transitioning from one state to the other (and vice-versa). Several previous works have proposed that approximating the eigenvectors of the graph Laplacian can be useful in RL. For example, Mahadevan (2005) shows that using the eigenvectors as basis functions can accelerate learning with policy iteration. Machado et al. (2017a;b) show that the eigenvectors can be used to construct options with exploratory behavior. The Laplacian eigenvectors are also a natural solution to the aforementioned reward-shaping problem. If we use a uniformly random behavior policy, the Laplacian state representations will be appropriately aware of the walls present in the gridworld and will induce an L2 distance as shown in Figure 1(right). This choice of representation accurately reflects the geometry of the problem, not only providing a strong learning signal at every state, but also avoiding spurious local optima.

While the potential benefits of using Laplacian-based representations in RL are clear, current techniques for approximating or learning the representations are ill-suited for model-free RL. For one, current methods mostly require an eigendecomposition of a matrix. When this matrix is the actual Laplacian (Mahadevan, 2005), the eigendecomposition can easily become prohibitively expensive. Even for methods which perform the eigendecomposition on a reduced matrix (Machado et al., 2017a;b), the eigendecomposition step may be computationally expensive, and furthermore precludes the applicability of the method to stochastic or online settings, which are common in RL. Perhaps more crucially, the justification for many of these methods is made in the tabular setting. The applicability of these methods to more general settings is unclear.

To resolve these limitations, we propose a computationally efficient approach to approximate the eigenvectors of the Laplacian with function approximation based on the spectral graph drawing objective, an objective whose optimum yields the desired eigenvector representations. We present the objective in a fully general RL setting and show how it may be stochastically optimized over minibatches of sampled experience. We empirically show that our method provides a better approximation to the Laplacian eigenvectors than previous proposals, especially when the raw representation is not tabular. We then apply our representation learning procedure to reward shaping in goal-achieving tasks, and show that our approach outperforms both sparse rewards and rewards based on L2 distance in the raw feature space. Results are shown under a set of gridworld maze environments and difficult continuous control navigation environments.

## 2 BACKGROUND

We present the eigendecomposition framework in terms of general Hilbert spaces. By working with Hilbert spaces, we provide a unified treatment of the Laplacian and our method for approximating its eigenvectors (Cayley, 1858) – *eigenfunctions* in Hilbert spaces (Riesz, 1910) – regardless of the underlying space (discrete or continuous). To simplify the exposition, the reader may substitute the following simplified definitions:

- The *state space* $S$ is a finite enumerated set $\{1, \ldots, |S|\}$.
- The *probability measure* $\rho$ is a probability distribution over $S$.
- The *Hilbert space* $\mathcal{H}$ is $\mathbb{R}^{|S|}$, for which elements $f \in \mathcal{H}$ are $|S|$ dimensional vectors representing *functions* $f : S \to \mathbb{R}$.
- The *inner product* $\langle f, g \rangle_{\mathcal{H}}$ of two elements $f, g \in \mathcal{H}$ is a weighted dot product of the corresponding vectors, with weighting given by $\rho$; i.e. $\langle f, g \rangle_{\mathcal{H}} = \sum_{u=1}^{|S|} f(u)g(u)\rho(u)$.

- A *linear operator* is a mapping $A : \mathcal{H} \to \mathcal{H}$ corresponding to a weighted matrix multiplication; i.e. $Af(u) = \sum_{v=1}^{|S|} f(v) A(u, v) \rho(v)$.
- A *self-adjoint* linear operator $A$ is one for which $\langle f, Ag \rangle_{\mathcal{H}} = \langle Af, g \rangle_{\mathcal{H}}$ for all $f, g \in \mathcal{H}$. This corresponds to $A$ being a symmetric matrix.

## 2.1 A SPACE AND A MEASURE

We now present the more general form of these definitions. Let $S$ be a set, $\Sigma$ be a $\sigma$-algebra, and $\rho$ be a measure such that $(S, \Sigma, \rho)$ constitutes a measure space. Consider the set of square-integrable real-valued functions $L^2(S, \Sigma, \rho) = \{ f : S \to \mathbb{R} \text{ s.t. } \int_S |f(u)|^2 \, d\rho(u) < \infty \}$. When associated with the inner-product,

$$\langle f, g \rangle_{\mathcal{H}} = \int_S f(u) g(u) \, d\rho(u),$$

this set of functions forms a complete inner product Hilbert space (Hilbert, 1906; Riesz, 1910). The inner product gives rise to a notion of orthogonality: Functions $f, g$ are orthogonal if $\langle f, g \rangle_{\mathcal{H}} = 0$. It also induces a norm on the space: $||f||^2 = \langle f, f \rangle_{\mathcal{H}}$. We denote $\mathcal{H} = L^2(S, \Sigma, \rho)$ and additionally restrict $\rho$ to be a probability measure, i.e. $\int_S 1 \, d\rho(u) = 1$.

## 2.2 THE LAPLACIAN

To construct the graph Laplacian in this general setting, we consider linear operators $D$ which are Hilbert-Schmidt integral operators (Bump, 1998), expressable as,

$$Df(u) = \int_S f(v) D(u, v) \, d\rho(v),$$

where with a slight abuse of notation we also use $D : S \times S \mapsto R^+$ to denote the kernel function. We assume that (i) the kernel function $D$ satisfies $D(u, v) = D(v, u)$ for all $u, v \in S$ so that the operator $D$ is self-adjoint; (ii) for each $u \in S$, $D(u, v)$ is the Radon-Nikodym derivative (density function) from some probability measure to $\rho$, i.e. $\int_S D(u, v) \, d\rho(v) = 1$ for all $u$. With these assumptions, $D$ is a compact, self-adjoint linear operator, and hence many of the spectral properties associated with standard symmetric matrices extend to $D$.

The Laplacian $L$ of $D$ is defined as the linear operator on $\mathcal{H}$ given by,

$$Lf(u) = f(u) - \int_S f(v) D(u, v) \, d\rho(v) = f(u) - Df(u). \tag{1}$$

The Laplacian may also be written as the linear operator $I - D$, where $I$ is the identity operator. Any eigenfunction with associated eigenvalue $\lambda$ of the Laplacian is an eigenfunction with eigenvalue $1 - \lambda$ for $D$, and vice-versa.

**Our goal** is to find the first $d$ eigenfunctions $f_1, ..., f_d$ associated with the smallest $d$ eigenvalues of $L$ (subject to rotation of the basis).[1] The mapping $\phi : S \mapsto \mathbb{R}^d$ defined by $\phi(u) = [f_1(u), ..., f_d(u)]$ then defines an embedding or representation of the space $S$.

## 2.3 SPECTRAL GRAPH DRAWING

Spectral graph drawing (Koren, 2003) provides an optimization perspective on finding the eigenvectors of the Laplacian. Suppose we have a large graph, composed of (possibly infinitely many) vertices with weighted edges representing pairwise (non-negative) affinities (denoted by $D(u, v) \geq 0$ for vertices $u$ and $v$). To visualize the graph, we would like to embed each vertex in a low dimensional space (e.g., $\mathbb{R}^d$ in this work) so that pairwise distances in the low dimensional space are small for vertices with high affinity. Using our notation, the graph drawing objective is to find a set of orthonormal functions $f_1, \ldots, f_d$ defined on the space $S$ which minimize

$$G(f_1, \ldots, f_d) = \frac{1}{2} \int_S \int_S \sum_{k=1}^d (f_k(u) - f_k(v))^2 D(u, v) \, d\rho(u) \, d\rho(v). \tag{2}$$

---

[1]The existence of these eigenfunctions is formally discussed in Appendix A.

The orthonormal constraints can be written as $\langle f_j, f_k \rangle_{\mathcal{H}} = \delta_{jk}$ for all $j, k \in [1, d]$ where $\delta_{jk} = 1$ if $j = k$ and $\delta_{jk} = 0$ otherwise.

The graph drawing objective (2) may be expressed more succinctly in terms of the Laplacian:

$$G(f_1, \ldots, f_d) = \sum_{k=1}^{d} \langle f_k, L f_k \rangle_{\mathcal{H}}. \tag{3}$$

The minimum value of (3) is the sum of the $d$ smallest eigenvalues of $L$. Accordingly, the minimum is achieved when $f_1, \ldots, f_d$ span the same subspace as the corresponding $d$ eigenfunctions. In the next section, we will show that the graph drawing objective is amenable to stochastic optimization, thus providing a general, scalable approach to approximating the eigenfunctions of the Laplacian.

## 3 REPRESENTATION LEARNING WITH THE LAPLACIAN

In this section, we specify the meaning of the Laplacian in the RL setting (i.e., how to set $\rho, D$ appropriately). We then elaborate on how to approximate the eigenfunctions of the Laplacian by optimizing the graph drawing objective via stochastic gradient descent on sampled states and pairs of states.

### 3.1 THE LAPLACIAN IN A REINFORCEMENT LEARNING SETTING

In RL, an agent interacts with an environment by observing states and acting on the environment. We consider the standard MDP setting (Puterman, 1990). Briefly, at time $t$ the environment produces an observation $s_t \in S$, which at time $t = 0$ is determined by a random sample from an environment-specific initial distribution $P_0$. The agent's policy produces a probability distribution over possible actions $\pi(a|s_t)$ from which it samples a specific action $a_t \in A$ to act on the environment. The environment then yields a reward $r_t$ sampled from an environment-specific reward distribution function $R(s_t, a_t)$, and transitions to a subsequent state $s_{t+1}$ sampled from an environment-specific transition distribution function $P(s_t, a_t)$. We consider defining the Laplacian with respect to a fixed behavior policy $\pi$. Then, the transition distributions $P^{\pi}(s_{t+1}|s_t)$ form a Markov chain. We assume this Markov chain has a unique stationary distribution.

We now introduce a choice of $\rho$ and $D$ for the Laplacian in the RL setting. We define $\rho$ to be the stationary distribution of the Markov chain $P^{\pi}$ such that for any measurable $U \subset S$ we have $\rho(U) = \int_S P^{\pi}(U|v) \, d\rho(v)$.

As $D(u, v)$ represents the pairwise affinity between two vertices $u$ and $v$ on the graph, it is natural to define $D(u, v)$ in terms of the transition distribution.[2] Recall that $D$ needs to satisfy (i) $D(u, v) = D(v, u)$ (ii) $D(u, \cdot)$ is the density function from a probability measure to $\rho$ for all $u$. We define

$$D(u, v) = \frac{1}{2} \left. \frac{dP^{\pi}(s|u)}{d\rho(s)} \right|_{s=v} + \frac{1}{2} \left. \frac{dP^{\pi}(s|v)}{d\rho(s)} \right|_{s=u}, \tag{4}$$

which satisfies these conditions[3]. In other words, the affinity between states $u$ and $v$ is the average of the two-way transition probabilities: If $S$ is finite then the first term in (4) is $P^{\pi}(s_{t+1} = v|s_t = u)/\rho(v)$ and the second term is $P^{\pi}(s_{t+1} = u|s_t = v)/\rho(u)$.

### 3.2 APPROXIMATING THE LAPLACIAN EIGENFUNCTIONS

Given this definition of the Laplacian, we now aim to learn the eigen-decomposition embedding $\phi$. In the model-free RL context, we have access to states and pairs of states (or sequences of states) only via sampling; i.e. we may sample states $u$ from $\rho(u)$ and pairs of $u, v$ from $\rho(u)P^{\pi}(v|u)$. This imposes several challenges on computing the eigendecomposition:

- Enumerating the state space $S$ may be intractable due to the large cardinality or continuity.
- For arbitrary pairs of states $(u, v)$, we do not have explicit access to $D(u, v)$.
- Enforcing exact orthonormality of $f_1, ..., f_d$ may be intractable in innumerable state spaces.

---

[2]The one-step transitions can be generalized to multi-step transitions in the definition of $D$, which provide better performance for RL applications in our experiments. See Appendix B for details.

[3]$D(u, v) = D(v, u)$ follows from definition. See Appendix B for a proof that $D(u, \cdot)$ is a density.

With our choices for $\rho$ and $D$, the graph drawing objective (Eq. 2) is a good start for resolving these challenges because it can be expressed as an expectation (see Appendix C for the derivation):

$$G(f_1, \ldots, f_d) = \frac{1}{2} \mathbb{E}_{u \sim \rho, v \sim P^\pi(\cdot|u)} \Big[ \sum_{k=1}^d (f_k(u) - f_k(v))^2 \Big]. \tag{5}$$

Minimizing the objective with stochastic gradient descent is straightforward by sampling transition pairs $(s_t, s_{t+1})$ as $(u, v)$ from the replay buffer. The difficult part is ensuring orthonormality of the functions. To tackle this issue, we first relax the orthonormality constraint to a soft constraint $\sum_{j,k} (\langle f_j, f_k \rangle_{\mathcal{H}} - \delta_{jk})^2 < \epsilon$. Using standard properties of expectations, we rewrite the inequality as follows:

$$\sum_{j,k} \left( \mathbb{E}_{u \sim \rho} \left[ f_j(u) f_k(u) \right] - \delta_{jk} \right)^2 = \sum_{j,k} \left( \mathbb{E}_{u \sim \rho} \left[ f_j(u) f_k(u) - \delta_{jk} \right] \right)^2$$

$$= \sum_{j,k} \mathbb{E}_{u \sim \rho} \left[ f_j(u) f_k(u) - \delta_{jk} \right] \mathbb{E}_{v \sim \rho} \left[ f_j(v) f_k(v) - \delta_{jk} \right]$$

$$= \sum_{j,k} \mathbb{E}_{u \sim \rho, v \sim \rho} \left[ \left( f_j(u) f_k(u) - \delta_{jk} \right) \left( f_j(v) f_k(v) - \delta_{jk} \right) \right] < \epsilon.$$

In practice, we transform this constraint into a penalty and solve the unconstrained minimization problem. The resulting penalized graph drawing objective is

$$\tilde{G}(f_1, \ldots, f_d) = G(f_1, \ldots, f_d) + \beta \mathbb{E}_{u \sim \rho, v \sim \rho} \Big[ \sum_{j,k} \left( f_j(u) f_k(u) - \delta_{jk} \right) \left( f_j(v) f_k(v) - \delta_{jk} \right) \Big], \tag{6}$$

where $\beta$ is the penalty weight (KKT multiplier).

The $d$-dimensional embedding $\phi(u) = [f_1(u), ..., f_d(u)]$ may be learned using a neural network function approximator. We note that $\tilde{G}$ has a form which appears in many other representation learning objectives, being comprised of an attractive and a repulsive term. The attractive term minimizes the squared distance of embeddings of randomly sampled transitions experienced by the policy $\pi$, while the repulsive term repels the embeddings of states independently sampled from $\rho$. The repulsive term is especially interesting and we are unaware of anything similar to it in other representation learning objectives: It may be interpreted as orthogonalizing the embeddings of two randomly sampled states while regularizing their norm away from zero by noticing

$$\sum_{j,k} \left( f_j(u) f_k(u) - \delta_{jk} \right) \left( f_j(v) f_k(v) - \delta_{jk} \right) = \left( \phi(u)^T \phi(v) \right)^2 - \|\phi(u)\|_2^2 - \|\phi(v)\|_2^2 + d. \tag{7}$$

## 4 RELATED WORK

One of the main contributions of our work is a principled treatment of the Laplacian in a general RL setting. While several previous works have proposed the use of the Laplacian in RL (Mahadevan, 2005; Machado et al., 2017a), they have focused on the simple, tabular setting. In contrast, we provide a framework for Laplacian representation learning that applies generally (i.e., when the state space is innumerable and may only be accessed via sampling).

Our main result is showing that the graph drawing objective may be used to stochastically optimize a representation module which approximates the Laplacian eigenfunctions. Although a large body of work exists regarding stochastic approximation of an eigendecomposition (Cardot & Degras, 2018; Oja, 1985), many of these approaches require storage of the entire eigendecomposition. This scales poorly and fails to satisfy the desiderata for model-free RL – a function approximator which yields arbitrary rows of the eigendecomposition. Some works have proposed extensions that avoid this requirement by use of Oja's rule (Oja, 1982). Originally defined within the Hebbian framework, recent work has applied the rule to kernelized PCA (Xie et al., 2015), and extending it to settings similar to ours is a potential avenue for future work. Other approaches to eigendecomposition that may eventually prove fruitful in the RL setting include Mall et al. (2013), which proposes to scale to large datasets by subsampling representative subgraphs, and Alzate & Suykens (2010), which provides some techniques to extend spectral clustering to out-of-sample points.

In RL, Machado et al. (2017b) propose a method to approximate the Laplacian eigenvectors with functions approximators via an equivalence between proto-value functions (Mahadevan, 2005) and spectral decomposition of the successor representation (Stachenfeld et al., 2014). Importantly, they propose an approach for stochastically approximating the eigendecomposition when the state space is large. Unfortunately, their approach is only justified in the tabular setting and, as we show in our results below, does not generalize beyond. Moreover, their eigenvectors are based on an explicit eigendecomposition of a constructed reduced matrix, and thus are not appropriate for online settings.

Approaches more similar to ours (Shaham et al., 2018; Pfau et al., 2018) optimize objectives similar to Eq. 2, but handle the orthonormality constraint differently. Shaham et al. (2018) introduce a special-purpose orthonormalizing layer, which ensures orthonormality at the mini-batch level. Unfortunately, this does not ensure orthonormality over the entire dataset and requires large mini-batches for stability. Furthermore, the orthonormalization process can be numerically unstable, and in our preliminary experiments we found that TensorFlow frequently crashed due to numerical errors from this sort of orthonormalization. Pfau et al. (2018) turn the problem into an unconstrained optimization objective. However, in their chosen form, one cannot compute unbiased stochastic gradients. Moreover, their approach scales quadratically in the number of embedding dimensions. Our approach does not suffer from these issues.

Finally, we note that our work provides a convincing application of Laplacian representations on difficult RL tasks, namely reward-shaping in continuous-control environments. Although previous works have presented interesting preliminary results, their applications were either restricted to small discrete state spaces (Mahadevan, 2005) or focused on qualitative assessments of the learned options (Machado et al., 2017a;b).

## 5 EXPERIMENTS

### 5.1 EVALUATING THE LEARNED REPRESENTATIONS

We first evaluate the learned representations by how well they approximate the subspace spanned by the smallest eigenfunctions of the Laplacian. We use the following evaluation protocol: (i) Given an embedding $\phi : S \to \mathbb{R}^d$, we first find its principal $d$-dimensional orthonormal basis $h_1, ..., h_d$, onto which we project all embeddings in order to satisfy the orthonormality constraint of the graph drawing objective; (ii) the evaluation metric is then computed as the value of the graph drawing objective using the projected embeddings. In this subsection, we use finite state spaces, so step (i) can be performed by SVD.

We used a FourRoom gridworld environment (Figure 2). We generate a dataset of experience by randomly sampling $n$ transitions using a uniformly random policy with random initial state. We compare the embedding learned by our approximate graph drawing objective against methods proposed by Machado et al. (2017a;b). Machado et al. (2017a) find the first $d$ eigenvectors of the Laplacian by eigen-decomposing a matrix formed by stacked transitions, while Machado et al. (2017b) eigen-decompose a matrix formed by stacked learned successor representations. We evaluate the methods with three different raw state representations of the gridworld: (i) one-hot vectors ("index"), (ii) $(x, y)$ coordinates ("position") and (iii) top-down pixel representation ("image").

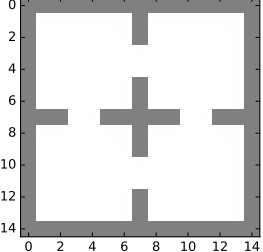

Figure 2: FourRoom Env.

We present the results of our evaluations in Figure 3. Our method outperforms the previous methods with all three raw representations. Both of the previous methods were justified in the tabular setting, however, surprisingly, they underperform our method even with the tabular representation. Moreover, our method performs well even when the number of training samples is small.

### 5.2 LAPLACIAN REPRESENTATION LEARNING FOR REWARD SHAPING

We now move on to demonstrating the power of our learned representations to improve the performance of an RL agent. We focus on a family of tasks – *goal-achieving* tasks – in which the agent is rewarded for reaching a certain state. We show that in such settings our learned representations are well-suited for reward shaping.

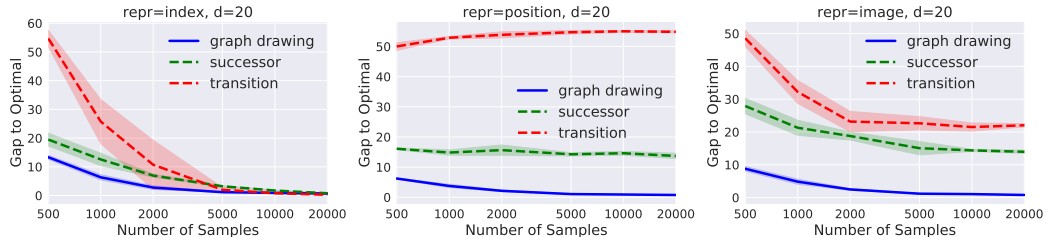

Figure 3: Evaluation of learned representations. The x-axis shows number of transitions used for training and y-axis shows the gap between the graph drawing objective of the learned representations and the optimal Laplacian-based representations (lower is better). We find our method (graph drawing) more accurately approximates the desired representations than previous methods. See Appendix D for details and additional results.

**Goal-achieving tasks and reward shaping.** A goal-achieving task is defined by an environment with transition dynamics but no reward, together with a goal vector $z_g \in Z$, where $Z$ is the goal space. We assume that there is a known predefined function $h : S \to Z$ that maps any state $s \in S$ to a goal vector $h(s) \in Z$. The learning objective is to train a policy that controls the agent to get to some state $s$ such that $\|h(s) - z_g\| \leq \epsilon$. For example the goal space may be the same as the state space with $Z = S$ and $h(s) = s$ being the identity mapping, in which case the target is a state vector. More generally the goal space can be a subspace of the state space. For example, in control tasks a state vector may contain both position and velocity information while a goal vector may just be a specific position. See Plappert et al. (2018) for an extensive discussion and additional examples.

A reward function needs to be defined in order to apply reinforcement learning to train an agent that can perform a goal achieving task. Two typical ways of defining a reward function for this family of tasks are (i) the sparse reward: $r_t = -1[\|h(s_{t+1}) - z_g\| > \epsilon]$ as used by Andrychowicz et al. (2017) and (ii) the shaped reward based on Euclidean distance $r_t = -\|h(s_{t+1}) - z_g\|$ as used by Pong et al. (2018); Nachum et al. (2018). The sparse reward is consistent with what the agent is supposed to do but may slow down learning. The shaped reward may either accelerate or hurt the learning process depending on the whether distances in the raw feature space accurately reflect the geometry of the environment dynamics.

**Reward shaping with learned representations.** We expect that distance based reward shaping with our learned representations can speed up learning compared to sparse reward while avoiding the bias in the raw feature space. More specifically, we define the reward based on distance in a learned latent space. If the goal space is the same as the state space, i.e. $S = Z$, the reward function can be defined as $r_t = -\|\phi(s_{t+1}) - \phi(z_g)\|$. If $S \neq Z$ we propose two options: (i) The first is to learn an embedding $\phi : Z \to \mathbb{R}^d$ of the goal space and define $r_t = -\|\phi(h(s_{t+1})) - \phi(z_g)\|$. (ii) The second options is to learn an an embedding $\phi : S \to \mathbb{R}^d$ of the state space and define $r_t = -\|\phi(s_{t+1}) - \phi(h^{-1}(z_g))\|$, where $h^{-1}(z)$ is defined as picking arbitrary state $s$ (may not be unique) that achieves $h(s) = z$. We experiment with both options when $S \neq Z$.

### 5.2.1 GRIDWORLD

We experiment with the gridworld environments with $(x, y)$ coordinates as the observation. We evaluate on three different mazes: OneRoom, TwoRooms and HardMaze, as shown in the top row of Figure 4. The red grids are the goals and the heatmap shows the distances from each grid to the goal in the learned Laplacian embedding space. We can qualitatively see that the learned rewards are well-suited to the task and appropriately reflect the environment dynamics, especially in TwoRoom and HardMaze where the raw feature space is very ill-suited.

These representations are learned according to our method using a uniformly random behavior policy. Then we define the shaped reward as a half-half mix of the L2 distance in the learned latent space and the sparse reward. We found this mix to be advantageous, as the L2 distance on its own does not provide enough difference between the reward of the goal state and rewards of the states near the goal. When the L2 distance between the representations of the goal state and adjacent states is small the Q-function can fail to provide a significant signal to actually reach the goal state (rather than a state that is just close to the goal). Thus, to better align the shaped reward with the task directive, we use a half-half mix, which clearly draws a boundary between the goal state and its adjacent states (as the sparse reward does) while retaining the structure of the distance-shaped reward. We plot the

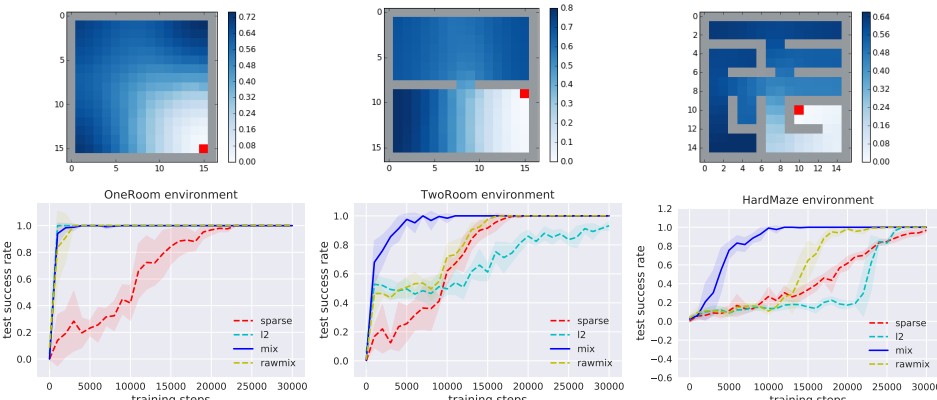

Figure 4: Results of reward shaping with a learned Laplacian embedding in GridWorld environments. The top row shows the L2 distance in the learned embedding space. The bottom row shows empirical performance. Our method (mix) can reach optimal performance faster than the baselines, especially in harder mazes. Policies are trained by DQN (Mnih et al., 2013).

learning performance of an agent trained according to this learned reward in Figure 4. All plots are based on 5 different random seeds. We compare against (i) **sparse**: the sparse reward, (ii) **l2**: the shaped reward based on the L2 distance in the raw $(x, y)$ feature space, (iii) **rawmix**: the mixture of (i) and (ii). Our mixture of shaped reward based on learning representations and the sparse reward is labelled as **"mix"** in the plots. We observe that in the OneRoom environment all shaped reward functions significantly outperform the sparse reward, which indicates that in goal-achieving tasks properly shaped reward can accelerate learning of the policy, justifying our motivation of applying learned representations for reward shaping. In TwoRoom and HardMaze environments when the raw feature space cannot reflect an accurate distance, our Laplacian-based shaped reward learned using the graph drawing objective ("mix") significantly outperforms all other reward settings.

### 5.2.2 CONTINUOUS CONTROL

To further verify the benefit of our learned representations in reward shaping, we also experiment with continuous control navigation tasks. These tasks are much harder to solve than the gridworld tasks because the agent must simultaneously learn to control itself and navigate to the goal. We use Mujoco (Todorov et al., 2012) to create 3D mazes and learn to control two types of agents, PointMass and Ant, to navigate to a certain area in the maze, as shown in Figure 5. Unlike the gridworld environments the $(x, y)$ goal space is distinct from the state space, so we apply our two introduced methods to align the spaces: (i) learning $\phi$ to only embed the $(x, y)$ coordinates of the state (**mix**) or (ii) learning $\phi$ to embed the full state (**fullmix**). We run experiments with both methods. As shown in Figure 5 both "mix" and "fullmix" outperform all other methods, which further justifies the benefits of using our learned representations for reward shaping. It is interesting to see that both embedding the goal space and embedding the state space still provide a significant advantage even if neither of them is a perfect solution. For goal space embedding, part of the state vector (e.g. velocities) is ignored so the learned embedding may not be able to capture the full structure of the environment dynamics. For state space embedding, constructing the state vector from the goal vector makes achieving the goal more challenging since there is a larger set of states (e.g. with different velocities) that achieve the goal but the shaped reward encourage the policy to reach only one of them. Having a better way to align the two spaces would be an interesting future direction.

## 6 CONCLUSION

We have presented an approach to learning a Laplacian-based state representation in RL settings. Our approach is both general – being applicable to any state space regardless of cardinality – and scalable – relying only on the ability to sample mini-batches of states and pairs of states. We have further provided an application of our method to reward shaping in both discrete spaces and continuous-control settings. With our scalable and general approach, many more potential applications of Laplacian-based representations are now within reach, and we encourage future work to continue investigating this promising direction.

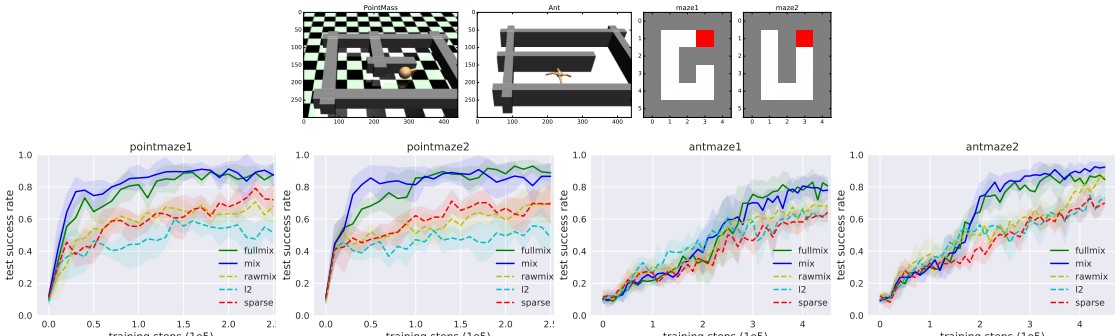

Figure 5: Results of reward shaping with a learned Laplacian embedding in continuous control environments. Our learned representations are used by the "mix" and "fullmix" variants (see text for details), whose performance dominates that of all other methods. Policies are trained by DDPG (Lillicrap et al., 2015).

ACKNOWLEDGMENTS

We thank Marc Bellemare, Dale Schuurmans, and the Google Brain team for insightful comments and discussions.

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

## A    EXISTENCE OF SMALLEST EIGENVALUES OF THE LAPLACIAN.

Since the Hilbert space $\mathcal{H}$ may have infinitely many dimensions we need to make sure that the smallest $d$ eigenvalues of the Laplacian operator is well defined. Since $L = I - D$ if $\lambda$ is an eigenvalue of $D$ then $1 - \lambda$ is an eigenvalue of $L$. So we turn to discuss the existence of the largest $d$ eigenvalues of $D$. According to our definition $D$ is a compact self-adjoint linear operator on $\mathcal{H}$. So it has the following properties according to the spectral theorem:

- $D$ has either (i) a finite set of eigenvalues or (ii) countably many eigenvalues $\{\lambda_1, \lambda_2, ...\}$ and $\lambda_n \to 0$ if there are infinitely many. All eigenvalues are real.
- Any eigenvalue $\lambda$ satisfies $-\|D\| \le \lambda \le \|D\|$ where $\|\cdot\|$ is the operator norm.

If the operator $D$ has a finite set of $n$ eigenvalues its largest $d$ eigenvalues exist when $d$ is smaller than $n$.

If $D$ has a infinite but countable set of eigenvalues we first characterize what the eigenvalues look like:

Let $f_1$ be $f_1(u) = 1$ for all $u \in S$. Then $Df_1(u) = \int f_1(v)D(u,v)\,d\rho(v) = \int D(u,v)\,d\rho(v) = 1$ for all $u \in S$ thus $Df_1 = f_1$. So $\lambda = 1$ is an eigenvalue of $D$.

Recall that the operator norm is defined as

$$\|D\| = \inf\left\{c \ge 0 : \|Df\| \le c\|f\|\,\forall f \in \mathcal{H}\right\}.$$

Define $q_u$ be the probability measure such that $\frac{dq_u}{d\rho} = D(u,\cdot)$. We have

$$Df(u)^2 = \mathbb{E}_{v \sim q_u}[f(v)]^2 \le \mathbb{E}_{v \sim q_u}[f(v)^2] = \int f(v)^2 D(u,v)\,d\rho(v)$$

and

$$\|Df\|^2 = \int Df(u)^2\,d\rho(u) \le \int\int f(v)^2 D(u,v)\,d\rho(v)\,d\rho(u) = \int f(v)^2\,d\rho(v) = \|f\|^2,$$

which hold for any $f \in \mathcal{H}$. Hence $\|D\| \le 1$.

So the absolute values of the eigenvalues of $D$ can be written as a non-increasing sequence which converges to $0$ with the largest eigenvalue to be $1$. If $d$ is smaller than the number of positive eigenvalues of $D$ then the largest $d$ eigenvalues are guaranteed to exist. Note that this condition for $d$ is stricter than the condition when $D$ has finitely many eigenvalues. We conjecture that this restriction is due to an artifact of the analysis and in practice using any value of $d$ would be valid when $\mathcal{H}$ has infinite dimensions.

## B    DEFINING $D$ FOR MULTI-STEP TRANSITIONS

To introduce a more general definition of $D$, we first introduce a generalized discounted transition distribution $P_\lambda^\pi$ defined by

$$P_\lambda^\pi(v|u) = \sum_{\tau=1}^\infty (\lambda^{\tau-1} - \lambda^\tau)P^\pi(s_{t+\tau} = v|s_t = u), \tag{8}$$

where $\lambda \in [0, 1)$ is a discount factor, with $\lambda = 0$ corresponding to the one-step transition distribution $P_0^\pi = P^\pi$. Notice that $P_\lambda^\pi(v|u)$ can be also written as $P_\lambda^\pi(v|u) = \mathbb{E}_{\tau \sim q_\lambda}\left[P^\pi(s_{t+\tau} = v|s_t = u)\right]$ where $q_\lambda(\tau) = \lambda^{\tau-1} - \lambda^\tau$. So sampling from $P_\lambda^\pi(v|u)$ can be done by first sampling $\tau \sim q_\lambda$ then rolling out the Markov chain for $\tau$ steps starting from $u$.

Note that for readability we state the definition of $P_\lambda^\pi$ in terms of discrete probability distributions but in general $P_\lambda^\pi(\cdot|u)$ are defined as a probability measure by stating the discounted sum (8) for any measurable set of states $U \in \Sigma, U \subset S$ instead of a single state $v$.

Also notice that when $\lambda > 0$ sampling $v$ from $P_\lambda^\pi(v|u)$ required rolling out more than one steps from $u$ (and can be arbitrarily long). Given that the replay buffer contains finite length (say $T'$)

trajectories sampling exactly from the defined distribution is impossible. In practice, after sampling $u = s_t$ in a trajectory and $\tau$ from $q_\lambda(\tau)$ we discard this sample if $t + \tau > T'$.

With the discounted transition, distributions now the generalized $D$ is defined as

$$
D(u, v) = \frac{1}{2} \left. \frac{dP_\lambda^\pi(s|u)}{d\rho(s)} \right|_{s=v} + \frac{1}{2} \left. \frac{dP_\lambda^\pi(s|v)}{d\rho(s)} \right|_{s=u}. \tag{9}
$$

We assume that $P_\lambda^\pi(\cdot|u)$ is absolutely continuous to $\rho$ for any $u$ so that the Radon Nikodym derivatives are well defined. This assumption is mild since it is saying that for any state $v$ that is reachable from some state $u$ under $P^\pi$ we have a positive probability to sample it from $\rho$, i.e. the behavior policy $\pi$ is able to explore the whole state space (not necessarily efficiently).

**Proof of $D(u, \cdot)$ being a density of some probability measure with respect to $\rho$.** We need to show that $\int_S D(u, v)\, d\rho(v) = 1$. Let $f(\cdot|u)$ be the density function of $P_\lambda^\pi(\cdot|u)$ with respect to $\rho$ then $D(u, v) = \frac{1}{2} f(v|u) + \frac{1}{2} f(u|v)$. According to the definition of $f$ we have $\int_S f(v|u)\, d\rho(v) = 1$. It remains to show that $\int_S f(u|v)\, d\rho(v) = 1$ for any $u$.

First notice that if $\rho$ is the stationary distribution of $P^\pi$ it is also the stationary distribution of $P_\lambda^\pi$ such that $\rho(U) = \int_S P^\pi(U|v)\, d\rho(v)$ for any measurable $U \subset S$. Let $g(u) = \int_S f(u|v)\, d\rho(v)$. For any measurable set $U \subset S$ we have

$$
\begin{aligned}
\int_U g(u)\, d\rho(u) &= \int_{u \in U} \int_{v \in S} f(u|v)\, d\rho(v)\, d\rho(u) \\
&= \int_{v \in S} \int_{u \in U} f(u|v)\, d\rho(u)\, d\rho(v) \\
&= \int_{v \in S} P_\lambda^\pi(U|v)\, d\rho(v) \\
&= \rho(U), \qquad\qquad \text{(Property of the stationary distribution.)}
\end{aligned}
$$

which means that $g$ is the density function of $\rho$ with respect to $\rho$. So $g(u) = 1$ holds for all $u$. (For simplicity we ignore the statement of "almost surely" throughout the paper.)

**Discussion of finite time horizon.** Because proving $D(u, \cdot)$ to be a density requires the fact that $\rho$ is the stationary distribution of $P^\pi$, the astute reader may suspect that sampling from the replay buffer will differ from the stationary distribution when the initial state distribution is highly concentrated, the mixing rate is slow, and the time horizon is short. In this case, one can adjust the definition of the transition probabilities to better reflect what is happening in practice: Define a new transition distribution by adding a small probability to "reset": $\tilde{P}^\pi(\cdot|u) = (1 - \delta) P^\pi(\cdot|u) + \delta P_0$. This introduces a randomized termination to approximate termination of trajectories (e.g,. due to time limit) without adding dependencies on $t$ (to retain the Markov property). Then, $\rho$ and $D$ can be defined in the same way with respect to $\tilde{P}^\pi$. Now the replay buffer can be viewed as rolling out a single long trajectory with $\tilde{P}^\pi$ so that sampling from the replay buffer approximates sampling from the stationary distribution. Note that under the new definition of $D$, minimizing the graph drawing objective requires sampling state pairs that may span over the "reset" transition. In practice, we ignore these pairs as we do not want to view "resets" as edges in RL. When $\delta$ (e.g. $1/T$) is small, the chance of sampling these "reset" pairs is very small, so our adjusted definition still approximately reflects what is being done in practice.

## C   DERIVATION OF (5)

$$G(f_1, \ldots, f_d) = \frac{1}{2} \int_S \int_S \sum_{k=1}^{d} (f_k(u) - f_k(v))^2 D(u,v) \, d\rho(u) \, d\rho(v)$$

$$= \frac{1}{4} \int_S \int_S \sum_{k=1}^{d} (f_k(u) - f_k(v))^2 \, dP_\lambda^\pi(v|u) \, d\rho(u)$$

$$+ \frac{1}{4} \int_S \int_S \sum_{k=1}^{d} (f_k(u) - f_k(v))^2 \, dP_\lambda^\pi(u|v) \, d\rho(v)$$

(Switching the notation $u$ and $v$ in the second term gives the same quantity as the first term)

$$= \frac{1}{2} \int_S \int_S \sum_{k=1}^{d} (f_k(u) - f_k(v))^2 \, dP_\lambda^\pi(v|u) \, d\rho(u)$$

$$= \frac{1}{2} \mathbb{E}_{u \sim \rho, v \sim P^\pi(\cdot|u)} \left[ \sum_{k=1}^{d} (f_k(u) - f_k(v))^2 \right].$$

## D   ADDITIONAL RESULTS AND EXPERIMENT DETAILS

### D.1   EVALUATING THE LEARNED REPRESENTATIONS

**Environment details.** The FourRoom gridworld environment, as shown in Figure 2, has 152 discrete states and 4 actions. A tabular (index) state representation is a one hot vector with 152 dimensions. A position state representation is a two dimensional vector representing the $(x, y)$ coordinates, scaled within $[-1, 1]$. A image state representation contains 15 by 15 RGB pixels with different colors representing the agent, walls and open ground. The transitions are deterministic. Each episode starts from a uniformly sampled state has a length of 50. We use this data to perform representation learning and evaluate the final learned representation using our evaluation protocol.

**Implementation of baselines.** Both of the approaches in Machado et al. (2017a) and Machado et al. (2017b) output $d$ *eigenoptions* $e_1, ..., e_d \in \mathbb{R}^m$ where $m$ is the dimension of a state feature space $\psi : S \to \mathbb{R}^m$, which can be either the raw representation (Machado et al., 2017a) or a representation learned by a forward prediction model (Machado et al., 2017b). Given the $d$ eigenoptions, an embedding can be obtained by letting $f_i(s) = \psi(s)^T e_i$. Following their theoretical results it can be seen that if $\psi$ is the one-hot representation of the tabular states and the stacked rows contains unique enumeration of all transitions/states $\phi = [f_1, ..., f_d]$ spans the same subspace as the smallest $d$ eigenvectors of the Laplacian.

**Additional results.** Additional results for $d = 50, 100$ are shown in Figure 6.

**Choice of $\beta$ in (6).** When the optimization problem associated with our objective (6) may be solved exactly, increasing $\beta$ will always lead to better approximations of the exact graph drawing objective (2) as the soft constraint approaches to the hard constraint. However, the optimization problem becomes harder to be solve by SGD when the value of $\beta$ is too large. We perform an sensitivity study over $\beta$ to show this trade-off in Figure 7. We can see that the optimal value of $\beta$ increases as $d$ in creases.

**Hyperparameters.** $D$ is defined using one-step transitions ($\lambda = 0$ in (9)). We use $\beta = d/20$, batch size 32, Adam optimizer with learning rate 0.001 and total training steps 100, 000. For representation mappings: we use a linear mapping for index states, a $200 \to 200$ two hidden layer fully connected neural network for position states and a convolutional network for image states. All activation functions are relu. The convolutional network contains 3 conv-layers with output channels $(16, 16, 16)$, kernel sizes $(4, 4, 4)$, strides $(2, 2, 1)$ and a final linear mapping to representations.

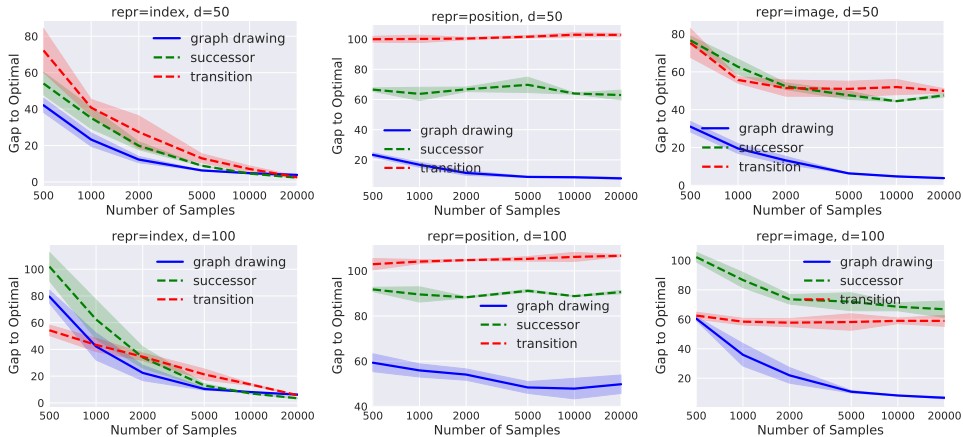

Figure 6: Evaluation of learned representations for $d = 50, 100$.

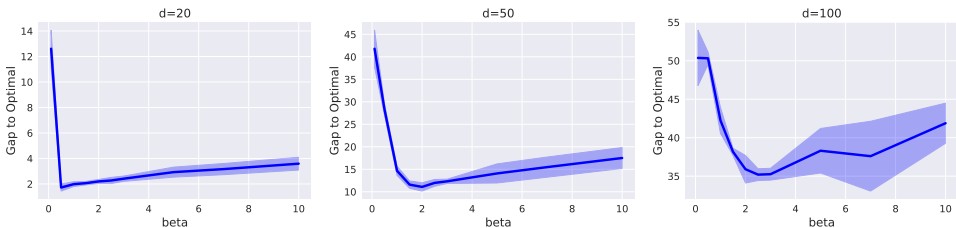

Figure 7: Ablation study for the value of $\beta$. We use $n = 2000$ and repr = position.

### D.2 LAPLACIAN REPRESENTATION LEARNING FOR REWARD SHAPING

#### D.2.1 GRIDWORLD

**Environment details** All mazes have a total size of 15 by 15 grids, with 4 actions and total number of states decided by the walls. We use $(x, y)$ position as raw state representations. Since the states are discrete the success criteria is set as reaching the exact grid. Each episode has a length of 50.

**Hyperparameters** For representation learning we use $d = 20$. In the definition of $D$ we use the discounted multi-step transitions (9) with $\lambda = 0.9$. For the approximate graph drawing objective (6) we use $\beta = 5.0$ and $\delta_{jk} = 0.05$ (instead of 1) if $j = k$ otherwise 0 to control the scale of L2 distances. We pretrain the representations for 30000 steps (This number of steps is not optimized and we observe that the training converges much earlier) by Adam with batch size 128 and learning rate 0.001. For policy training, we use the vanilla DQN (Mnih et al., 2013) with a online network and a target network both representing the Q-function. The policy used for testing is to select the action with the highest Q-value according to the online network at each state. The online network is trained to minimize the Bellman error by sampling transitions from the replay buffer. The target network is updated every 50 steps with a mixing rate of 0.05 (of the current online network with 0.95 of the previous target network). Epsilon greedy with $\epsilon = 0.2$ is used for exploration. Reward discount is 0.98. The policy is trained with Adam optimizer with learning rate 0.001. For both representation mapping and Q-functions we use a fully connected network (parameter not shared) with 3 hidden layers and 256 units in each layer. All activation functions are relu.

#### D.2.2 CONTINUOUS CONTROL

**Environment details** The PointMass agent has a 6 dimensional state space a 2 dimensional action space. The Ant agent has a 29 dimensional state space and a 8 dimensional action space. The success criteria is set as reaching an L2 ball centered around a specific $(x, y)$ position with the radius as 10% of the total size of the maze, as shown in Figure 5. Each episode has a length of 300.

**Hyperparameters** For representation learning we use $d = 20$. In the definition of $D$ we use the discounted multi-step transitions (9) with $\lambda = 0.99$ for PointMass and $\lambda = 0.999$ for Ant. For the approximate graph drawing objective (6) we use $\beta = 2.0$ and $\delta_{jk} = 0.1$ (instead of 1) if $j = k$ otherwise 0 to control the scale of L2 distances. We pretrain the representations for 50000 steps by Adam with batch size 128 and learning rate 0.001 for PointMass and 0.0001 for Ant. For policy training, we use the vanilla DDPG (Lillicrap et al., 2015) with a online network and a target network. Each network contains two sub-networks: an actor network representing the policy and a critic network representing the Q-function. The online critic network is trained to minimize the Bellman error and the online actor network is trained to maximize the Q-value achieved by the policy. The target network is updated every 1 step with a mixing rate of 0.001. For exploration we follow the Ornstein-Uhlenbeck process as described in the original DDPG paper (Lillicrap et al., 2015). Reward discount is 0.995. The policy is trained with Adam optimizer with batch size 100, actor learning rate 0.0001 and critic learning rate 0.001 for PointMass and 0.0001 for Ant. For representation mapping we use a fully connected network (parameter not shared) with 3 hidden layers and 256 units in each layer. Both actor network and critic network have 2 hidden layers with units $(400, 300)$. All activation functions are relu.

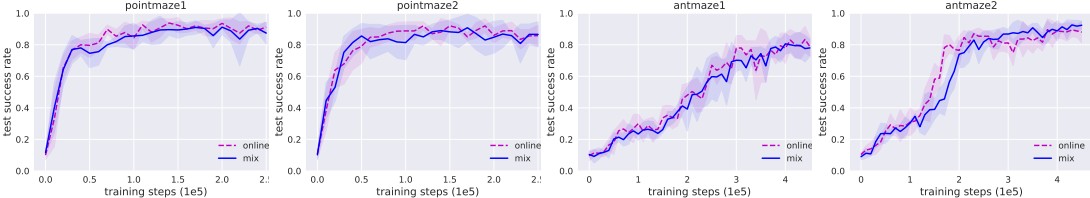

Figure 8: Results of reward shaping with pretrained-then-fixed v.s. online-learned representations.

**Online learning of representations** We also present results of learning the representations online instead of pretraining-and-fix and observe equivalent performance, as shown in Figure 8, suggesting that our method may be successfully used in online settings. For online training the agent moves faster in the maze during policy learning so we anneal the $\lambda$ in $D$ from its inital value to 0.95 towards the end of training with linear decay. The reason that online training provides no benefit is that our randomized starting position setting enables efficient exploration even with just random walk policies. Investigating the benefit of online training in exploration-hard tasks would be an interesting future direction.

