# OpenReview forum: "The Laplacian in RL: Learning Representations with Efficient Approximations"
_ICLR.cc/2019/Conference_

### Official Review · AnonReviewer1 · 2018-10-30
**The Laplacian in RL: Learning Representations with Efficient Approximations**

**Rating:** 7
**Confidence:** 4

**Review:**

The authors propose a Laplacian in the context of reinforcement learning, together with learning the representations. Overall the authors make a nice contribution. The insight of defining rho to be the stationary distribution of the Markov chain P^pi and connecting this to eq (1) is interesting. Also the definition of the reward function on p.7 in terms of the distance between phi(s_{t+1}) and phi(z_g) looks original. The method is also well illustrated and compared with other methods, showing the efficiency of the proposed method.

On the other hand I also have further comments and suggestions:

- it would be good if the authors could comment on the choice of d. This is in fact a model selection problem. According to which criterion is this selected?

- the authors define D(u,v) in eq (4). Why this choice? Is there some intuition or interpretation possible related to this expression?

- in (6) beta is called a Lagrange multiplier. Given that a soft constraint (not a hard constraint) is added for the orthonormality constraint it is not a Lagrange multiplier.

How sensitive are the results with respect to the choice of beta in (6) (or epsilon in the eq above)? The orthonormality constraint will only be approximately satisfied. Isn't this a problem?

Wouldn't it be better in this case to rely on optimization algorithm on Grassmann and Stiefel manifolds?

- The authors provide a scalable approach related to section 2 by stochastic optimization. Other scalable methods related to kernel spectral clustering (related to subsets/subgraphs and making out-of-sample extensions) were proposed in literature, e.g.

Multiway Spectral Clustering with Out-of-Sample Extensions through Weighted Kernel PCA, IEEE Transactions on Pattern Analysis and Machine Intelligence, 32(2), 335-347, 2010.

Kernel Spectral Clustering for Big Data Networks, Entropy, Special Issue: Big Data, 15(5), 1567-1586, 2013.

---

> ### Author Response · Authors · 2018-11-17
> **Response**
>
> We thank the reviewer for the careful reading of the paper. We are glad the reviewer found the contribution of the paper insightful and original.  Responses to the reviewer’s questions are below:
>
> “it would be good if the authors could comment on the choice of d. This is in fact a model selection problem. According to which criterion is this selected?”
>
> -- Our choice of d(=20) in reward shaping experiments is arbitrary and we didn’t tune it. In practice, if the downstream task is known, d can be regarded as a hyperparameter and selected according to the performance. If the downstreaming task is not available, one can visualize the distances between representations like in Figure 4 (with randomly sampled goal states) and select d when the visualized distance is meaningful; or in other cases treat it as an additional hyperparameter to search over.
>
>
> “the authors define D(u,v) in eq (4). Why this choice? Is there some intuition or interpretation possible related to this expression?”
>
> -- The underlying motivation is in order to make the graph drawing objective practical to optimize (via sampling) while reflecting the affinity between states. Optimizing the graph drawing objective requires sampling from D(u,v)rho(u)rho(v) so D(u,v)rho(u)rho(v) should be a joint measure over u, v. The Laplacian is defined for undirected graphs so D(u,v) also needs to be symmetric. These are the intuitions behind the conditions for D in Section 2.2. In RL, a natural choice for representing the affinity between two states is to use the transition probabilities P(u|v) (which is also convenient for sampling).  However, naively setting D := P is premature, as P in general does not satisfy the conditions necessary for D.  To this end, we first “symmetrize” P to achieve the setting of D as in Eq 4 by averaging the transitions u->v and v->u  This procedure is analogous to “symmetrized Laplacians” (see Boley, et al., “Commute times for a directed graph using an asymmetric Laplacian”). We then divide it by rho to make D(u,v)rho(u)rho(v) a joint measure over pairs of states so that the graph drawing objective can be written in terms of an expectation (as in (5)) and sample based optimization is possible.
>
>
> “in (6) beta is called a Lagrange multiplier. Given that a soft constraint (not a hard constraint) is added for the orthonormality constraint it is not a Lagrange multiplier.”
>
> -- We have updated the paper to replace this terminology with the more appropriate “KKT multiplier”.
>
>
> “How sensitive are the results with respect to the choice of beta in (6) (or epsilon in the eq above)? The orthonormality constraint will only be approximately satisfied. Isn't this a problem?”
>
> -- The results are not very sensitive to the choice of beta. We have plots for approximation qualities with different values of beta in Appendix D-1 Figure-7 with discussions.
> -- Approximately satisfying the orthonormality constraint is not a problem in RL applications, at least in the reward shaping setting which we experiment with. In reward shaping the important thing is that the distance in the latent space can reflect the affinity between states properly, and orthonormality constraint plays a role more like encouraging the diversity of the representations (preventing them from collapsing to a single point). We think the same argument applies to most other applications of learned representations to RL so only satisfying the constraint approximately should not be a problem in the RL context.
>
>
> “Wouldn't it be better in this case to rely on optimization algorithm on Grassmann and Stiefel manifolds?”
>
> -- In the RL setting, one requires an optimization algorithm which is amenable to stochastic mini-batching. We are not aware of an optimization algorithm based on Grassman and Stiefel manifolds which is applicable in such settings, but would be interested if the reviewer has a specific algorithm in mind.  While our paper proposes one technique for enforcing orthonormality, there are likely to be other applicable algorithms to achieve the same aims, and we would be happy to include references to them as alternative methods.
>
>
> “Other scalable methods related to kernel spectral clustering (related to subsets/subgraphs and making out-of-sample extensions) were proposed in literature”
>
> -- We updated our paper to cite these two papers in the related work section.

---

### Official Review · AnonReviewer3 · 2018-11-02
**needs improvement**

**Rating:** 7
**Confidence:** 3

**Review:**

Summary: This paper proposes a method to learn a state representation for RL using the Laplacian. The proposed method aims to generalize previous work, which has only been shown in finite state spaces, to continuous and large state spaces. It goes to approximate the eigenvectors of the Laplacian which is constructed using a uniformly random policy to collect training data. One use-case of the learnt state representation is for reward-shaping that is said to accelerate the training of standard goal-driven RL algorithms.


In overall, the paper is well written and easy to follow. The idea that formulates the problem of approximating the Laplacian engenfunctions as constraint optimization is interesting. I have some following major concerns regarding to the quality and presentation of the paper.

- Though the idea of learning a state representation seems interesting and might be of interest within the RL research, the authors have not yet articulated the usefulness of this learnt representation. For larger domains, learning such a representation using a random policy might not be ideal because the random policy can not explore the whole state space efficiently. I wish to see more discussions on this, e.g. transfer learning, multi-task learning etc.

- In terms of an application of the learnt representation, reward-shaping looks interesting and promising. However I am concerned about its sample efficiency and comparing experiments. It takes a substantial amount of data generated from a random policy to attain such a reward-shaping function, so the comparisons in Fig.5 are not fair any more in terms of sample efficiency. On the other hand, the learnt representation for reward-shaping is fixed to one goal, can one do transfer learning/multi-task learning to gain the benefit of such an expensive step of representation learning with a random policy.

- The second equation, below the text"we rewrite the inequality as follows" in page 5, is correct? this derivation is like E(X^2) = E(X) E(X)?

- About the performance reported in Section 5.1, I wonder if the gap can be closer to zero if more eigenfunctions are used?


================
After rebuttal:
Thanks the authors for clarification. I have read the author's responses to my review. The authors have sufficiently addressed my concerns. I agree with the responses and decide to change my overall rating

---

> ### Author Response · Authors · 2018-11-17
> **Response**
>
> We thank the reviewer for the valuable feedback. We are glad the reviewer found the paper interesting and easy to follow.  Responses to the reviewer’s remaining concerns are addressed below.  With these, we hope the reviewer will find the paper more appropriate for publication and, if so, will raise their score accordingly.  We are also always happy to discuss further if the reviewer has additional concerns.
>
> “learning such a representation using a random policy might not be ideal because the random policy can not explore the whole state space efficiently”
>
> -- We agree that this can be a concern. However, a random policy can be sufficient for exploration when the initial state is uniformly sampled from the whole state space (as we did in our experiments). As you suggest, a random policy is not sufficient for exploration when the initial state is not sampled from the whole state space but only sampled within a region that is far from the goal. In this case, exploring the whole state space itself is a hard problem which we are not trying to solve here. In this paper, we aim at demonstrating the usefulness of learned representations in “reward shaping” with well controlled experiments in RL settings, so we attempted to exclude other factors such as exploration.
> -- With that being said, we have results showing that our representation learning method works beyond random-walk policies: In appendix D-2 we have experiments (Figure-8) showing that the learned representation with online policies provides a similar advantage in reward shaping as with random-walk policies. Here, the online policy and the representation are learned concurrently starting from scratch and on the same online data.  It is thus significant that we retain the same advantages in speed of training.
>
>
> “I am concerned about its sample efficiency and comparing experiments”
>
> -- Even when the pretraining samples are included, our method is much more sample efficient than the baselines. The representation learning phase with a random walk policy is not expensive. For the MuJoCo experiments in Figure 5, we pretrain the representation with 50,000 samples.Then, we train the policy with 250,000(for pointmass)/450,000(for ant) samples.  After shifting the mix/fullmix learning curves to the right by 50,000 steps to include the pretraining samples, their learning curves are still clearly above the baseline learning curves.
>
>
> “the learnt representation for reward-shaping is fixed to one goal, can one do transfer learning/multi-task learning to gain the benefit of such an expensive step of representation learning with a random policy”
>
> - Our learnt representation is not fixed to one goal and are in fact agnostic to goal or task reward. Thus, the representations may be used for any goals in subsequent training. The goal is used only when computing the rewards (L2 distances) for training goal-achieving policies.
> - The representation learning phase is not expensive compared with the policy training phase, as we explained in the previous concern point.
>  - The representations are learned in a purely unsupervised way without any task information (e.g. goal, reward, a good policy). So it is natural to apply the representations to different tasks without the notion of “transfer” or “multi-task”.
>
>
> “The second equation, below the text "we rewrite the inequality as follows" in page 5, is correct?”
>
> -- Yes, it is correct. The square is outside the brackets in all of the expressions, so E(X)^2 = E(X)E(X).
>
>
> “About the performance reported in Section 5.1, I wonder if the gap can be closer to zero if more eigenfunctions are used?”
>
> -- We have additional results for larger values of d (50, 100) in Appendix D-1, Figure 6. The gap actually becomes bigger if more eigenfunctions are used: With much larger values of d the problem becomes harder as you need to approximate (the subspace of) more eigenfunctions of the Laplacian.

---

### Official Review · AnonReviewer2 · 2018-11-03
**well written, interesting approach, well evaluated**

**Rating:** 7
**Confidence:** 3

**Review:**

This works proposes a scalable way of approximating the eigenvectors of the Laplacian in RL by optimizing the graph drawing objective on limited sampled states and pairs of states. The authors empirically show the benefits of their method in two different types of goal achieving task.

Pros:
- Well written, well structured, an overall enjoyable read.
- The related work section appears to be comprehensive and supports the motivations for the presented work.
- Clear and rigorous derivations.
- The method is evaluated both in terms of how well it is able to approximate the optimal Laplacian-based representations with limited samples compared to baseline models and how well it solves reward shaping in RL.

Cons:
- In the experimental section, the methods used to learn the policies, DQN and DDPG, should be briefly explained or at least referenced.
- A further discussion on why the authors chose a half-half mix of the L2 distance and sparse reward could be beneficial. The provided explanation (L2 distance doesn't provide enough gradient) is not very convincing nor justified.

---

> ### Author Response · Authors · 2018-11-17
> **Response**
>
> We are glad that the reviewer found the paper interesting, well-written, and well-evaluated.  We also appreciate the feedback.
>
> With regards to the methods DQN and DDPG, we have updated the paper to include references in the main text and brief descriptions of these algorithms in the experiment details section in Appendix.
>
> We have updated the paper to clarify the reasoning behind the half-half mix for reward shaping. By “gradient,”  we meant the change in rewards between adjacent states (not the gradient in optimization). When the L2 distance between the representations of the goal state and adjacent states is small the Q-function can fail to provide a significant signal to actually reach the goal state (rather than a state that is just close to the goal).  Thus, to better align the shaped reward with the task directive, we use a half-half mix, which clearly draws a boundary between the goal state and its adjacent states (as the sparse reward does) while retaining the structure of the distance-shaped reward.

---

### Meta-Review · Area_Chair1 · 2018-12-13
**Well-written paper and a useful extension to approximating the eigenvectors of the Laplacian**

**Confidence:** 4
**Recommendation:** Accept (Poster)

**Metareview:**

This paper provides a novel and non-trivial method for approximating the eigenvectors of the Laplacian, in large or continuous state environments. Eigenvectors of the Laplacian have been used for proto-value functions and eigenoptions, but it has remained an open problem to extend their use to the non-tabular case. This paper makes an important advance towards this goal, and will be of interest to many that would like to learn state representations based on the geometric information given by the Laplacian.

The paper could be made stronger by including a short discussion on why the limitations of this approach. Its an important new direction, but there must still be open questions (e.g., issues with the approach used to approximate the orthogonality constraint). It will be beneficial to readers to understand these issues.